# Diagnosis in Neuroendocrine Neoplasms: From Molecular Biology to Molecular Imaging

**DOI:** 10.3390/cancers14102514

**Published:** 2022-05-19

**Authors:** Ray Manneh Kopp, Paula Espinosa-Olarte, Teresa Alonso-Gordoa

**Affiliations:** 1Sociedad de Oncología y Hematología del Cesar, Valledupar 200001, Colombia; 2Hospital General Universitario de Valencia, 46014 Valencia, Spain; espinosa_pau@gva.es; 3Hospital Universitario Ramón y Cajal, 28034 Madrid, Spain; talonsogordoa@gmail.com

**Keywords:** neuroendocrine, diagnosis, molecular biology

## Abstract

**Simple Summary:**

Neuroendocrine neoplasms are a small group of malignancies with a diverse prognosis and behaviour. In order to offer an adequate treatment, physicians need to perform a proper diagnosis, staging and stratification. This review aims to help to integrate the information from pathology, immunohistochemistry, molecular biology and imaging to guide this process.

**Abstract:**

Neuroendocrine neoplasms (NENs) are a heterogeneous group of tumours with a diverse behaviour, biology and prognosis, whose incidence is gradually increasing. Their diagnosis is challenging and a multidisciplinary approach is often required. The combination of pathology, molecular biomarkers, and the use of novel imaging techniques leads to an accurate diagnosis and a better treatment approach. To determine the functionality of the tumour, somatostatin receptor expression, differentiation, and primary tumour origin are the main determining tumour-dependent factors to guide treatment, both in local and metastatic stages. Until recently, little was known about the biological behaviour of these tumours. However, in recent years, many advances have been achieved in the molecular characterization and diagnosis of NENs. The incorporation of novel radiotracer-based imaging techniques, such as ^68^Gallium-DOTATATE PET-CT, has significantly increased diagnostic sensitivity, while introducing the theragnosis concept, offering new treatment strategies. Here, we will review current knowledge and novelties in the diagnosis of NENs, including molecular biology, pathology, and new radiotracers.

## 1. Introduction

Neuroendocrine neoplasms (NENs) are one of the most challenging groups of tumours to deal with in oncology. They include a heterogeneous spectrum of tumours, mainly originating in the GI tract, lungs, and pancreas. Their increasing incidence and prevalence, their different clinical behavior, including hormonal manifestations, and the availability of new drugs and diagnosis techniques, has led to a pathology of exciting management that integrates different disciplines around the patients dealing with this disease.

In this review, we aim to analyze the key diagnostic procedures concerning the pathological and molecular features of NEN, as well as the best imaging approach, including conventional and molecular functional imaging. The best therapeutic decisions for our patients can then be made based on this information.

## 2. Clinical Presentation

Neuroendocrine neoplasms can retain the distinctive feature of producing hormones or biogenic amines that are responsible for particular clinical syndromes, giving them the label of “functioning-tumours”. They represent around 20% of well-differentiated NENs. Clinical suspicion is necessary to perform the diagnosis approach and to choose the best symptomatic treatment. The most frequent is the carcinoid syndrome, but many others have been identified, as summarised in Table 1 [1,2,3]. In non-functioning tumours, the clinical manifestations are secondary to tumour growth, which depends on tumour location. On one hand, they can go unnoticed for years and incidentally diagnosed in an image technique or surgery conducted for other reasons. On the other hand, poorly differentiated NENs or neuroendocrine carcinomas (NECs) are most often presented with a constitutional syndrome (asthenia, anorexia, weight loss) as other non-neuroendocrine malignancies.

Key messages:-NENs are a very heterogeneous group of tumours with many different clinical presentations. Diagnosis of functioning NETs is a challenge and clinical suspicion is crucial.-Carcinoid syndrome is the most frequent hormonal syndrome. It is characterized mainly by diarrhea and flushing, but life-threatening manifestations such as carcinoid heart disease or carcinoid crisis could appear.

NECs presentation used to be similar to other non-neuroendocrine malignancies.
cancers-14-02514-t001_Table 1Table 1Main syndromes and clinical manifestations of functioning-NENs.SyndromeFrequency *Association with Hereditary sd.Cell of OriginMost Frequent Tumour LocationClinical FeaturesHormoneCARCINOID SD.19% of NETs--Enterochromaffin cellsSmall intestine > bronchial > pancreaticDiarrhea, flushing, sweating, hypotension, bronchospasm, carcinoid heart disease, carcinoid crisisSerotonin > histamine, brady-tachykinis, kallikrein, prostaglandins (atypical sd.)GASTRINOMAZollinger-Ellison Sd.4% of NETs25% associated with MEN-1G cellsDuodenum > pancreas > other: thymusRefractory peptic ulcers or in atypical locations. Chronic diarrhea that responds to PPIsGastrinINSULINOMA8% of NETs6–7% associated with MEN-1Pancreatic β cellsPancreasHypoglycemia: headache, lethargy, blurred vision, tremor, palpitations, sweating, anxiety, behavioral disorders, seizures, etc.InsulinVIPOMA0.8% of NETs5% associated withMEN-1VIP producing cellsPancreas > extra-pancreatic tumoursSevere secretory diarrhea, dehydration, hypokalemia, hypochlorhydria, hypercalcaemia, hot flushes (20%)VIP GLUCAGONOMA1.5% of NETs5–17% associated with MEN-1Pacreatic α cellsPancreasWeight loss, diarrhea, MD, cheilitis, glossitis, necrolytic erythema migrans, hypercoagulabilityGlucagonSOMATOSTATINOMA0.1% of NETs--Pancreatic δ cellsPancreasMD, decreased gastric acid secretion, cholelithiasis, anemia, steatorrhea, and weight loss.SomatostatinTUMOUR MALIGNANT HYPERCALCEMIAUnfrequent--Ectopic secretion in tumoral cellsPancreasAnorexia, nausea, vomiting, abdominal painPTHrpACTHOMAUnfrequentRarely associated with MEN-1 y 2, MTC, VHLEctopic secretion in tumoral cellsForegut (larynx, thymus, lung, stomach, duodenum and pancreas), NECsSd. Ectopic or paraneoplastic Cushing: weight gain, DM, severe HTN, osteoporosis, muscle atrophy, hypokalemia, alkalosis, depression/psychosisACTHPPOMAUnfrequentRarely associated with MEN, VHLPP cellsPancreasInactive hormone, clinical mass effectPPGHRHOMAVery unfrequent--Ectopic secretion in tumoral cellsBronchial >Pancreas, SCNEC, pheochromocytoma, adrenalAcromegaly, gigantism, hyperinsulinemia, hypercortisolismGHRHECLOMAVery unfrequent--Enterochromaffin cellsStomachHistaminergic symptoms, gastric acid hypersecretionHistamineGHRELIOMAVery unfrequent--Gr cellsStomachMass effect. Orexigenic hormone, absence of cachexiaGrCCKOMAVery unfrequent--I cellsDuodenum, pancreasGastrinoma-like symptoms. Diarrhea, gastric ulcers, cholelithiasis, weight lossCCKGIPOMAVery unfrequent--K cellsDuodenum, jejunumIncreases insulin secretionGIPACTH, adrenocorticotropic hormone; CCK, cholecystokinin; GHRH, growth-hormone-releasing hormone; GIP, gastric inhibitory polypeptide; Gr, ghrelin; HTN; MD, Mellitus diabetes; MEN-1, multiple endocrine neoplasias type 1; MEN-2, multiple endocrine neoplasias type 2; MTC, Medullary thyroid cancer; NETs, neuroendocrine tumours; PTHrp, parathyroid hormone-related protein; SCNEC, small cell neuroendocrine carcinoma; Sd., syndrome; VHL, Von Hippel–Lindau; VIP, vasoactive intestinal peptide * Related to NETs. Data from RGETNE (Registry from Spanish group of NETs) [4].

## 3. Pathological Examination and Classification

The diagnosis of NENs must be conducted by a biopsy of the primary tumour or a metastatic site. In the lungs, bronchoscopy or transthoracic biopsy are the principal techniques, but mediastinoscopy or endobronchial endoscopic ultrasonography (EBUS) should be used if needed. Gastrointestinal NENs are normally diagnosed by upper or lower endoscopy while endoscopic ultrasonography (EUS) is mainly used in pancreatic lesions [5,6,7]. Sometimes, NETs are diagnosed after a surgical examination from a surgery performed for another reason. In such cases, the pathology results must be discussed by a multidisciplinary team and a complete oncological surgery should be considered.

Globally, NENs are divided into well differentiated, or NETs, poorly differentiated, or neuroendocrine carcinomas (NECs), and mixed histologies (mixed neuroendocrine and non-endocrine neoplasms, MiNENs, for GEP-NENs; or combined NEC and NSCLC for lung NENs). Classification is based on morphological differentiation and the proliferative index rate (either measured by Ki-67 or mitotic index in GEP origin or mitotic index in lung NENs) and the presence or absence of necrosis in lung NENs (Table 2 and Table 3).

Historically, Ki-67 proliferation index has not been determined in lung NENs, but there is increasing data regarding its utility in lung NENs. Thereby, determination has been included as “desirable” in the last 2021 WHO lung classification and recommended by other guidelines (ENETS, NANETS) [5,10,11]. Most typical carcinoids (TC) have a Ki-67 proliferation index < 5% and atypical carcinoids (AC) ≥ 5%, and some studies suggest that the NET-G3 group recognized for GEP NENs may also exist in lung NENs [12,13]. Determining Ki-67 can also be helpful in small samples where criteria for analyzing mitotic index cannot be met and for differential diagnosis between well and poorly differentiated NENs. Ki-67 could also be useful in NECs. The NORDIC NEC trial established the arbitrary threshold of 55% and showed that a Ki-67 > 55% was related to worse prognosis and platinum-based chemotherapy sensitivity. Nevertheless, this data is not validated [14].

NETs and NECs should be distinguished from other NEN, such as pheochromocitoma or paraganglioma, by demonstrating the epithelial origin of the tumour. For this purpose, the expression of cytokeratins must be assessed. Neuroendocrine cells express common classical markers, such as synaptophysin (more sensitive), chromogranin A (more specific) or CD56, whose determination is mandatory in order to confirm the neuroendocrine origin of the tumour. These markers are also present in neuronal cells, hereby called “neuro”-endocrine [15]. Many other inmunohistochemistry markers have been investigated, such as insulinoma associated protein 1 (INSM1), which appears to be a promising general neuroendocrine marker especially for the diagnosis of NECs [16].

Somatostatin receptors (STT) are overexpressed in more than 70–80% of NENs. The most frequently expressed is STT2, but up to five receptors have been described and their co-expression is common [17]. STT expression is higher in G1-2 NETs than in G3 NETs and NECs. STT expression is also related to the tumour origin, nearly always expressed in midgut in 80–90% of pancreatic NETs, and with a weak expression or non-expressed in lung NETs [18,19]. It can also be useful to distinguish between NET G3 and NECs, and of particular interest for SST-based image techniques and the use of targeted therapies, such as somatostatin analogues (Octeotride or Lanreotide), or radionuclides (177Lu-DOTATATE) [20,21,22].

Ten to twenty percent of metastatic NETs present as “unknown origin”. Both morphological features and immunohistochemistry can help to establish the primary tumour origin as this has prognosis and therapeutic implications.

Soga and Tazawa described four architectural patterns: nested (or type A), trabecular (B), pseudoglandular (C) and diffuse (D). Nested morphology is the most frequent and is usually, but not exclusively, present in ileal and jejunum NETs. Nevertheless, this is not conclusive as mixed patterns are frequent, especially in pancreatic and lung NETs [23]. More useful is the Iowa NET site of origin classifier, based on the presence of different neuroendocrine and epithelial markers. 90% of midgut NETs express CDX2 (caudal type homeobox 2), thereby it is proposed as the “first step” in the immunohistochemistry battery. CDX2 is also expressed in around 15% of pancreatic NETs with co-expression of pancreatic markers such as Islet 1 (ISL1) and PAX-6/8. Ninety-six percent of rectal NETs are SATB-2 (special AT-rich sequence-binding protein 2) positive, but it is also frequently expressed in appendiceal NETs (79%) [24]. For lung NETs, the 10% of midgut CDX 2-negative tumours, and the 5–10% of PAX6/ISL1 negative pancreatic NETs, other markers should be determined. OPT (orthopedia homeobox protein) is a neurodevelopmental transcription factor more sensitive and specific to lung carcinoids than the classical TTF-1 (thyroid transcription factor-1). It has also been suggested in combination with CD44 as a positive prognostic marker for lung carcinoids [25]. The addition of serotonin to CDX2 increases sensitivity to the diagnosis of midgut NET, from 90 to 96%, and should be considered as a second-step marker for those CDX-2 negative tumours [26]. A simplified tumour origin immunohistochemical algorithm has been proposed by Bellizzi after NET diagnosis (Figure 1) [27]. Immunohistochemistry for p53 and Rb can also be useful to distinguish G3 NETs from NECs. Biallellic inactivation of Rb and p53 are frequent in NECs (especially lung SCNECs) and they are rarely altered in NETs [5,6]. Other markers, such as DAXX/ATRX or clusterin, SSTR2, and CXCR4 can also be useful, as proposed by the University of Iowa, for morphologically ambiguous G3 neuroendocrine epithelial neoplasms.

Finally, to improve early tumour detection, prognostic assessment, and treatment monitoring, some approaches have been performed in liquid biopsy in NEN. Though different circulating biomarkers (chromogranin A and B, pancreastatin, neuron-specific enolase, pancreatic polypeptide) have been analyzed in NENs, they lack adequate accuracy or metrics to be considered in routine clinical practice as NET screening or monitoring markers, apart from specific hormones released in selected functioning tumours [28]. However, the NETest, based on a 51-gene expression by real-time polymerase chain reaction (RT-PCR) assay, has been evaluated in different NEN settings with promising results. As a diagnostic tool, NETest has demonstrated the highest diagnostic accuracy, of 95–96% [29], but interesting results have also been obtained in tumour recurrence prediction after local treatment and as a prognostic marker in advanced disease [30]. Moreover, the potential role of this RNA signature as a predictive marker of treatment response to optimizing NENs management in early progression detection is also under research [31].

Key messages:-For the diagnosis of NETs and NECs, the first step is to demonstrate the epithelial origin of the tumour by the expression of cytokeratins and the neuroendocrine nature by the expression of classical neuroendocrine markers (synaptophisin, chromogranin, and/or CD56);-Immunohistochemistry together with morphology are key to establishing the origin of the primary tumour;-The RNA signature NETest has shown promising results in optimizing the management of NENs as a diagnostic, prognostic and, potentially, predictive tool.

## 4. Molecular Alterations

Molecular alterations in NENs are mostly sporadic (95%), whereas around 5% belong to a hereditary syndrome. The identification of these germline alterations has helped to partially clarify some of the mechanisms involved in the pathogenesis of NENs. In the last years many advances have been made in the knowledge of the molecular biology of these tumours.

### 4.1. Hereditary Syndromes

The first molecular alterations known in NENs were those present in hereditary syndromes. At diagnosis, an exhaustive evaluation of familiar and personal medical history is required to identify NENs arising in the context of germline mutations in order to select the best treatment and appropriate follow-up. The main hereditary syndromes associated with NENs are summarized in Table 4 [32,33,34,35,36].

### 4.2. Sporadic Alterations

#### 4.2.1. Pancreatic NENs

As previously described, pancreatic NETs (P-NETs) can arise in the context of hereditary syndromes (*MEN1*, *VHL*, *NF1,* or *TSC*). The study of these families contributed to identifying pathogenic molecular aberrations in P-NETs. Nevertheless, recent studies have shown that the same genes can be altered in a variable percentage in sporadic P-NETs [37,38].

Compared to pancreatic ductal adenocarcinoma (PDAC), sporadic P-NETs show a lower mutational burden (0.82 vs. 2.64 mutations/Mb). Studies comparing PDACs and P-NETs also revealed that most frequently mutated genes in PDAC (*KRAS*, *TP53*, *TGF-**β*, *CDKN2A*) are not commonly modified in P-NETs, and their mutational pattern is also different (C: G vs. C: T in PDAC) [37,38,39].

The majority of sporadic P-NETs have somatic mutations in *MEN1* (37–44%) [37,38], that are present both in non-functioning (44%) and functioning tumours: glucagonoma (60%), VIPoma (57%), gastrinoma (38%) and insulinoma (2–19%) [33]. Mutually exclusive mutations in *DAXX* (death domain-associated protein) and *ATRX* (Alfa-talasemia or mental retardation syndrome X-linked) were initially described by Jiao et al. in 45% of P-NETs (*n* = 68) [37], and later validated in the whole genome sequencing study conducted by Scarpa et al. (*n* = 102) [38]. The inactivation of *DAXX*/*ATRX* complex has been related to chromosomal instability and correlated with alternative lengthening of telomeres (ALT), as a senescence scape mechanism induced by telomere shortening in many tumours [40]. Alterations in chromatin-remodeling genes, such as SETD2 (set-domain-containing 2), *MLL3* (myeloid/lymphoid or mixed-lineage leukemia protein 3), *ARID2* (AT-Rich Interaction Domain 2) and *SMARCA4* (SWI/SNF-related, matrix-associated, actin-dependent regulator of chromatin, subfamily a, member 4) were also identified by Scarpa et al. [38,41]. mTOR pathway mutations are present in 15% of sporadic P-NETs, including inactivating mutations of *PTEN*, *TSC1,* or *TSC2* and *DEPDC5*, and constitutive activating mutations in *PI2KCA* kinase domain [37,38].

The whole genome sequencing of 102 sporadic P-NETs conducted by Scarpa et al. showed germline mutations in 17% of the patients, including alterations in *MEN1* (6%), *CDKN1B* (1%), *VHL* (1%), and DNA repair machinery, such as *MUTYH1* (MuT Y homolog), *BRCA2,* and *CHEK2* (checkpoint kinase 2) [38]. In this study, somatic mutations were related to four principal pathways: chromatin-remodeling, DNA damage repair, activation of mTOR signaling, and telomere maintenance. It must be highlighted that *MEN1* is a common regulator of these four processes. The inactivation of *MEN1* leads to transcription deregulation secondary to histone modifications, the activation of *Akt-mTOR*, the suppression of DNA damage repair genes by homologous recombination, and the deregulation of telomerase reverse transcriptase (TERT). Gene expression profiles also unraveled a subgroup of tumours with aberrant expression of genes involved in *HIF* signaling [38].

Chromosomal rearrangements are less frequent in P-NETs than in PDAC. Recurrent deficiencies of chromosomes containing tumour suppressor genes involved in NETs tumorigenesis, such as loss of chr.11q (*MEN1*) or 9q (*CDKN2A*), have been described. Allelic losses (chr. 1, 3, 6, 8 and 10) or gains (chr. 4, 7, 12, 14, 17, 19 and 20) have also been reported [38,42]. Copy number variation pattern distinguished four groups of P-NETs: RPCL, recurrent pattern of whole chromosomal loss; limited copy number events, mainly losses in chr. 11; polyploidies; and aneuploidies [38].

Epigenetic mechanisms based on alteration of DNA methylation profiles seem to be of great interest in NETs biology. CpG island methylator phenotype, CIMP, has been described in around 83% of the P-NETs [43,44]. The main affected genes are *RASSF1A* (Ras association domain family 1 isoform A), involved in cellular cycle and apoptosis; *HIC-1* (hyper-methylated in cancer), a TP53 pathway effector; *APC* (*Adenomatous polyposis coli*), tumour suppressor gene responsible for familiar adenomatous polyposis; *CDKN2A*, suppressor gene encoding p16; *MGMT* (O-6-Methylguanine-DNA Methyltransferase), involved in DNA repair and likely with predictive value of response to alkylating agents such as temozolomide [45]; *MLH1* (MutL homology 1), related to microsatellites instability; and *VHL* [43,46,47].

*DAXX/ATRX* complex seems to play an important role in P-NETs methylation profile. Tumours harbouring *DAXX* or *ATRX* mutations have a different CIMP profile to non-mutated tumours [48]. De novo DNA methylations are carried out by methyltransferases DNMT3a and DNMT3b, and are maintained by DNMT1. *DAXX* closely interacts with DNMT1 and directs it to *RASSF1A* promoter [49].

Furthermore, *RB* and *TP53* are frequently altered in P-NECs, 74% and 95%, respectively. Loss of *RB* expression and *KRAS* mutations in P-NECs have been related to better response to platinum-based chemotherapy and can help to differentiate between P-NETs and P-NECs [50,51].

Key messages:-Sporadic P-NETs show a lower mutational burden and different mutated genes compared to classical PDAC.-Somatic mutations in MEN-1 and DAXX-ATRX are the most frequent in sporadic P-NETs; around 15% of sporadic P-NETs harbour mTOR pathway mutations. On the other hand, RB and TP 53 are the most frequently altered genes in P-NECs.-Germline mutations have been described in 17% of sporadic P-NETs, but further validation is needed.-Epigenetic mechanisms are of great interest in NETs; CpG island methylator phenotype has been described in around 83% of P-NETs.

#### 4.2.2. Small intestine NENs

Globally, small intestine NETs (SI-NETs) have a low mutational burden (0,77 mut/Mb) and a few genes with recurrent somatic mutations. The most frequently mutated gene is *CDKN1B,* in around 5–8% of SI-NETs. Recurrent altered genes are mainly due to CNVs; loss of 9,11,16 and, particularly chr. 18, and gains in chr. 4, 5, 14, 20 have been described [52,53,54,55]. Despite the frequent loss of heterozygosity (LOH) of chr. 18 (up to 78% of Si-NETs), and the efforts to identify suppressor genes located on this chromosome, the molecular mechanism by which LOH of chr.18 is involved in SI-NETs biology is not well understood [56,57]. The study from Karpathakis et al. (*n* = 97) showed that tumours with a LOH of chr. 18 have a different methylation profile than their non-tumoral counterpart. Moreover, they had 25 genes epigenetically deregulated, leading to a loss of expression of the encoding proteins. Among them, *LAMA3* (laminin alpha 3), involved in the genesis and function of the basement membrane, cellular migration, and signaling transduction; *SERPINB5* (serpin peptidase inhibitor clade B member *5*), member of *P53* pathway and involved in cell damage; and *RANK or TNFRSF11A* (tumour necrosis factor receptor superfamily member 11a NFKB activator), related to the *NF-**κB* pathway. They finally proposed three prognostic SI-NET groups based on their molecular profile: group A (55% of the patients), which included the 3 samples with mutations in *CDKN1B*, characterized by LOH of chr.18; group B (19%), with no large CNV; and group C (26%), characterized by multiple CNV and gains in chr. 4, 5, and 20. Median progression free survival (mPFS) after 10 years of follow-up was not reached in group A, and was about 56 and 21 months in groups B and C, respectively. These groups have a different methylation and expression profile, that mainly affected *EGFR*, *MAPK*, *Wnt*, *PI3K-mTOR,* and *VEGF* pathways [54]. Simbolo et al. identified frequent allelic loss in four genes located on chr. 18 (*BCL2, CDH19, DCC and SMAD4),* and loss in chr. 11 (38%) and chr. 16 (15%) in 44% of SI-NETs. This study showed that somatic mutations were infrequent in these tumours (34.6%) and confirmed previously described mutations in genes considered “drivers” in Si-NETs, such as *CDKN1B* (9.6%), *APC* (7.7%), and *BRAF* (3.8%) [53,54,58], but also in other genes not previously described in these neoplasms, such as *CDKN2C* (7.7%), *KRAS* (3.8%), *PIK3CA* (3.8%), and *TP53* (3.8%). They also reported that the amplification of *SRC* (v-src avian sarcoma (Schmidt-Ruppin A-2) viral oncogene homolog) located on chr.20, which shows copy number gains, was associated with worse prognosis. In 2020, the first genetically engineered mouse model of ileal NET was reported. Its development highlighted the loss of imprinting and subsequent upregulation of IGF2 in SI-NET formation, which was also demonstrated in 57% of SI-NET patients (*n* = 30). Thereby, IGF2 has been postulated as a driver gene in SI-NETs [59]. This study also revealed the key inactivation of *RB* and *P53* in the biology of these tumours. Although mutations in these genes are uncommon in SI-NETs, they could be inactivated by other mechanisms. Some studies have shown that *MIR1*, a Rb activator, located on chr.18, is downregulated in metastatic SI-NETs [60]; regarding *P53*, the overexpression of *MDM2*, a negative regulator of *P53* has been reported in SI-NETs [61].

Key messages:-SI-NETs show a low mutational burden and a few genes with recurrent somatic mutations, being the most frequent mutations in CDKN1B (around 5–8% of SI-NETs);-CNV are frequent in SI-NETs. LOH of chr. 18 has frequently been described although its meaning is not yet understood;-Although RB and TP53 are not frequently mutated in SI-NETs, they can be inactivated by other mechanisms.

#### 4.2.3. Lung NENs

SCLC or SCLNEC is the most studied NEN of the lungs. Globally, it is characterized by almost universal inactivation of TP53 and RB, the overexpression of oncogenes (Bcl-2 or MYC), and alteration in key proliferation pathways, such as PI3K/AKT/mTOR, which are constitutively activated, Hedgehog or Notch. CNV are common in SCLC; the loss in chr.3p is present in around 90% of SCLC and is considered to be present in pre-invasive lesions. Recently, Gay et al. described four SCLC subtypes characterized by the differential expression of three transcription factors, ASCL1, NEUROD1, and POU2F3, or low expression of all three with an inflamed gene signature (SCLC-A, *n*, P and I, respectively) with distinct response to systemic treatment [62,63].

In recent years, two independent groups (Fernandez-Cuesta et al. and Simbolo et al.) have contributed to increasing knowledge around the molecular biology of lung NENs using sequencing techniques in the four histology groups of lung NENs [64,65].

As in other NENs, lung carcinoids have a low mutational burden (0.4 mut/Mb vs. 10.5/Mb described in LCNECs). Moreover, NEC show tobacco-related mutational patterns (G:C > T:A), which are not present in lung carcinoids [12,64,65]. Despite lung carcinoids and NEC having some molecular alterations in common, their molecular profile is different. Mutations in *MEN1* are the most frequent in low-grade carcinoids, whereas mutations in *RB*, *TP53*, *PI3K/Akt/mTOR,* and *RAS* are more frequent in NECs.

*MEN1* in carcinoids (9–11%), is normally associated with a loss of heterozygosity of chr.11 [64,65]. MEN1 mutation implies loss of expression of its mRNA and has been related to worse prognosis [65,66]. The study conducted by Fernandez-Cuesta et al. included 44 paired samples (tumoral and non-tumoral) of lung carcinoids and identified mutations in *PSIP1* (PC4 And SFRS1 interacting protein 1), a gene involved in the same molecular pathway as *MEN1*, being mutually exclusive [64]. Germline mutations in *MEN1* are present in approximately 2% of lung carcinoid patients, in the context of *MEN1* hereditary syndrome.

Chromatin-remodeling genes are the most frequently mutated, both in NECs (55–70%) and lung carcinoids (40–52%) [12,64,65]. Among others, mutations are found in genes encoding histone methyltrasferases and demethylases, such as *SETDB1* (*SET domain bifurcated 1*), histone modifiers such as *BRWD3* (bromodomain and WE repeat containing 3), and *HDAC5* (*histone deacetilase* 5), genes of the Policomb complex and SWI/SNF complex such as *ARID1,* and *SMARC* (SWI/SNF related, matrix associated, actin dependent regulator of chromatin) family.

Simbolo et al. (*n* = 148) showed that CNV is rare in lung carcinoids and progressively increases in TC, AC, LCNEC, and SCNEC. Among NECs, there were frequent gains in chr. 5 (that contains the genes *TERT, SDHA, and RICTOR*) and chr.8, as well as LOH of chr.13 and chr.17p, where *RB1* and *TP53* are, respectively, located. Globally, *RB1* was the gene that most copy loss events showed (up to 91% of SCNEC) and was associated with worse prognosis, followed by *TP53*, which showed losses in one or both copies at up to 58.5% of SCNECs. These data have recently been validated in a different study conducted by the same group that included 67 samples of LCNECs and ACs [67].

Rekhtman et al. distinguished two groups of LCNECs based on their mutational profile (*n* = 45). Type I or *NSCLC-like,* characterized by mutations in *TP53* without co-mutations in *RB1*, and with molecular alterations typical of lung adenocarcinoma, such as mutations in *STK11, KRAS, KEAP1,* and *NFE2L2*. Type II or *SCLC-like* is characterized by co-mutation or allelic loss in both *RB1* and *TP53*. In the two groups, they identified frequent alterations in chromatin-remodeling genes, genes related with neurogenesis, DNA replication, and DNA repair [12]. In an independent cohort, LCNECs without alterations in *RB* responded better to chemotherapy used in NSCLC (platinum combined with taxanes or gemcitabine) than to the combination of platinum-etoposide frequently used in SCNEC. Thereby, the classification proposed by Rekhtman et al. could potentially help in treatment choice [68]. George et al. identified similar molecular groups in 75 samples of LCNEC (LCNEC type I and II). Nevertheless, although they showed similarities between genomic alterations in “LCNEC type I” and squamous or lung adenocarcinoma, their transcriptomic profile was closer to SCNEC, and, on the contrary, the group “LCNEC type II” with similar genetic alterations to SCNEC showed a different transcriptomic profile characterized by low expression of neuroendocrine biomarkers, high activity of NOTCH pathway, and a more immunogenic pattern [69]. Rekhtman et al. also identified two samples of LCNEC with a low mutational burden and inactivating mutations in *MEN1* that were included in a group called *carcinoid-like*.

Recently, Simbolo et al., in a cohort of LCNECs and ACs (*n* = 67), identified a 54-gene signature that defined three expression clusters with prognostic relevance [67]: the C1 enriched in LCNECs characterized by co-inactivation of *TP53* and *RB1*; C3, or enriched in ACs, with frequent mutations in *MEN1* and *TP53;* and C2, with 14 ACs and 8 LCNECs with mixed characteristics among both subgroups, with mutations in *TP53*, *MEN1* and *RB1*. In C3, the 83.3% were ACs, but this also included four LCNECs, three of them harbour mutations in *MEN1* and could correspond to the *carcinoid-like* group identified by Rekhtman et al. Median OS was not reached in C3, and was 19 and 47 months in C1 and C2, respectively.

In the multiomic work conducted by Alcala et al., which included 116 lung carcinoids (81 TCs and 35 ACs), 75 LCNECs and 66 SCNECs, three clusters with different clinical behaviour were described: “Carcinoid A” enriched in TCs, “Carcinoid B” enriched in ACs, and “LCNEC cluster”, with survival rates at ten years superior to 80%, 60% and 21%, respectively. Interestingly, in the “Carcinoid A” cluster, one LCNEC was included. This LCNEC showed similar molecular characteristics to TC, comparable to the group of *carcinoid-like* identified by Rekhman et al., so could correspond to “NET G3 GEP”, not yet recognized in lung NETs.

Apart from this possible new entity, Alcala et al. described a group of tumours called “*supracarcinoids*”. These tumours, although histologically classified as ACs, were included in the LCNEC cluster and had a more aggressive behaviour resembling LCNEC. This group made up 17% of all ACs. In the previous work of Simbolo et al., among the C2 cluster that comprised 14 AC and 8 LCNECs, two different subgroups were described. The C2a was similar to C1, enriched in LCNECs with a median Ki-67 of 60%, with frequent mutations in *TP53* and heterozygosity alterations in *RB1* [67]. The ACs included in the cluster C2a and C1 (13% of the AC) could correspond to the “*supracarcinoids*” described by Alcala et al.

One of the main controversies in lung NENs is whether there is a common origin of low and high grade tumours or, on the contrary, that they are a continuous entity and low grade tumours could dedifferentiate to highly poorly differentiated aggressive NECs. Pelosi et al. have recently made an in silico unsupervised classification in 148 lung NENs (including the four histologies) with exome data available. They classify 68% of the analysed samples in six histology-independent groups (C1-C6) and propose that NECs could develop from pre-existing carcinoids [70].

In summary, the molecular study of lung NENs has identified two possible new entities, “G3 NET”, a subgroup of LCNECs with less aggressive behaviour, resembling carcinoids, and one subgroup in the ACs called “supracarcinoids”, with a more aggressive behaviour. It has also revealed that LCNECs are a very heterogeneous group of tumours that can resemble LCNEC or SCNEC with prognostic and therapeutical implications. Contrary to the most widespread hypothesis that posits different origins of NETs and NECs, some molecular studies suggest that NETs could transform into NECs.

Key messages:-Lung carcinoids have a low mutational burden compared to LCNECs;-Chromatin-remodeling genes are the most frequently mutated, both in NECs and lung carcinoids;-Mutations in MEN1 are the most frequent in low-grade carcinoids, whereas mutations in RB, TP53, PI3K/Akt/mTOR, and RAS are more frequent in lung NECs;-Molecular study of lung NENs could potentially help to better classify these tumours, helping to better understand their biology and prognosis.

## 5. Imaging in the Diagnosis, Staging, and Follow up of Well-Differentiated Neuroendocrine Neoplasms

For diagnosis and staging, imaging is an important tool to guide therapeutic decisions in patients with localized, locally advanced, and metastatic disease. Surgery is the backbone of treatment of localized well-differentiated NENs. An adequate staging of patients is crucial to choosing the appropriate surgical approach. For that purpose, all these patients should be evaluated by an experienced multidisciplinary team.

Cross-sectional imaging is the most widely used approach to stage and follow patients with NENs. Generally, guidelines recommend multiphasic computed tomography (three-phase contrast-enhanced multi-slice CT) or magnetic resonance imaging (MRI) with contrast media [71,72].

Given the high chance of liver metastases in these neoplasms, it is recommended, when suspected, to perform MRI over CT and other modalities, since there is evidence showing an advantage in terms of sensitivity and specificity [73]. When a midgut NET origin is suspected, cross-sectional imaging (with intravenous, IV, and oral contrast), followed by endoscopy and video capsule endoscopy (when needed) are the most used tools to detect primary lesions, to elaborate a complete staging, and to provide information when performing a biopsy or a surgical procedure [74,75].

For pancreatic tumours, endoscopic ultrasonography (EUS) is an important tool and has a high sensitivity to detecting small lesions; this technique can improve the detection rates of pancreatic neuroendocrine tumours after other modalities have been attempted. A meta-analysis by James et al. showed that EUS identified pancreatic NETs in 97% of cases. One of the main benefits of EUS is the possibility of doing an EUS-guided fine needle biopsy for cytological diagnosis [7,76].

In the case of a NET of unknown primary, initial cross-sectional imaging is recommended; if there are inconclusive results, PET with 68Ga-DOTATOC, or OctreoSCAN with SPECT imaging when PET is not available, is the next step in the diagnosis algorithm. Colonoscopy and upper digestive endoscopy will also help to rule out tumours arising in the duodenum, distal ileum, and colon, as well as synchronic neoplasms.

When diagnosed in the metastatic setting, imaging of the brain is not usually required in asymptomatic patients. For the evaluation of liver metastases, multiphasic imaging with contrast enhancement should be performed, as well as chest CT scan with or without IV contrast. For the follow-up, imaging should be conducted 2–4 times a year based on tumour burden, pathologic characteristics, and symptoms.

### 5.1. Molecular Imaging

Somatostatin receptors (SSTRs), which are expressed in around 80% of NETs, are the main target of molecular imaging. They belong to the super-family of GPRC (G protein coupled receptor). There are five sub-types of SSTRs located in different chromosomes [77,78]. These receptors can be imaged, and their expression provides additional information in terms of treatment sensitivity/response, disease burden, extension, and location. The most widely used is a radiolabeled somatostatin analog (111-In petetreotide; OctreoScan), the first radiolabeled somatostatin analog approved for scintigraphy of NETs and useful in identifying primary tumours and sites of metastases. There are other radiolabeled analogs available, with different sensitivity and specificity: 99mTc (technetium)-labeled somatostatin analogs, such as 99mTC-depreotide and 99mTc-EDDA-HYNIC-TATE, which have shown some superiority compared to 11In-pentetreotide, in terms of sensitivity. Nowadays, 111In-Ocreotide SPECT/CT is recommended when SSTR-PET imaging is not available [7,79,80]. In Figure 2 is shown the structure and mechanism of action from somatostatin receptor imaging PET tracers.

#### 5.1.1. Positron Emission Tomography-Computed Tomography (PET-CT)

A major advance in molecular imaging for NETs is the use of Ga-68 DOTATATE. 68Ga is a trivalent radiometal that is relatively easy to produce and has a short half-life, decreasing the radiation exposure to the patient. There are several compounds available with different SSTR targets; for example, 68Ga-DOTATATE mainly targets SSTR 2 and 68Ga-DOTATOC SSTR 5. A systematic review and meta-analysis showed a pooled sensitivity of 91% (95% CI, 85–94%) and a pooled specificity of 94% (95% CI, 86–98%) for initial diagnosis [81,82].

The physiological distribution of 68Ga-DOTA-peptydes is higher in the spleen, adrenal glands, kidney, and pituitary glands, moderate in the liver, saliva glands, and thyroid, and it has a variable uptake in the stomach, colon, and the pancreatic activity of uncinate process. Besides NETs, there are other malignancies that may have a high positivity in 68Ga-SSTR-PET, such as medullary thyroid carcinoma, medulloblastoma, Merkel cell carcinoma, and meningioma.

Other tumour types, such as breast carcinoma, melanoma, lymphomas, non-small-cell lung cancer, sarcomas, prostate cancer, renal cell carcinoma, astrocytoma, and ependymoma, may be positive with a lower uptake [82,83]

Its higher sensitivity, and the morphological characterization with the CT, places this technique as the preferred option for initial diagnosis in the clinic.

According to consensus and guidelines [84], the main indications for SSTR PETs (68Ga-DOTATATE—68GaDODATOC—Cu-64 DOTATATE) are:

Initial diagnosis

-Initial staging after histopathologic diagnosis of NENs;-Staging NENs before planned surgery;-Location of primary tumour in patients with unknown primary;-Evaluation of a mass suggestive of NEN not accessible by biopsy;-Evaluation of patients with biochemical evidence and symptoms of NEN without tumour evidence on conventional imaging and without prior histologic diagnosis.-Follow-up and treatment-Restaging at clinical or biochemical progression without evidence from conventional imaging;-New indeterminate lesion on conventional imaging with unclear progression;-Monitoring of NENs seen predominantly on SSTR-PET;-Selection of patients for SSTR-targeted PRRT.

Baseline imaging with SSTR PETs is usually recommended for patients with advanced disease, in addition to conventional cross-sectional imaging. Both techniques allow clinicians to better determine the optimal treatment plan that should be determined by a multidisciplinary team with experience in the management of these neoplasms [85]. Such imaging tools add valuable information to tumour volume, burden of the disease, the Krenning score grade, or the functionality of the disease, among others. In fact, concerning functioning NENs, different radiotracers have been shown to increase sensitivity and specificity in different clinical syndromes, such as ectopic ACTH syndromes or insulinomas, due to their expression of glucagon-like peptide 1 receptor (GLP1R) [86,87]. Those functional imaging techniques improve the identification of the hormonal source or the diagnostic accuracy and are under continuous research to broaden their use in NENs [28].

In order to increase the detection rate of SSTR PETs, several trials have used intravenous contrasts showing a better detection rate of small liver lesions. A retrospective study of 66 patients with NENs evaluated the value of PET and triple-phase contrast-enhanced CT. They showed that 38% of patients had an impact in treatment decision after imaging [88]. The appropriate use criteria consensus guidance strongly recommends the use of IV contrast when possible [84]. Another way to improve the sensitivity and specificity of SSTR PETs is to combine it with MRI. PET/MRI imaging outperforms other techniques, especially in patients with predominant liver disease. A meta-analysis, including six studies, showed that 68Ga DOTA-SSA (included five trials with DOTA-TOC and one with DOTA-NOC) PET/MRI had a higher detection rate than PET-CT with a 15.3% added value over the latter [89].

Specifically for the detection of primary tumours in midgut NENs, several studies have evaluated the performance of 18F-fluorodihydroxyphenyl-L-alanine (18F-DOPA) compared with 68Ga DOTA-SSA. A meta-analysis that included six retrospective studies showed that 18F-DOPA PET-CT and 68Ga-DOTA peptides PET-CT identified the primary tumour in more than 80% of patients; of note, the authors showed a better primary lesion detection with 18F-DOPA PET-CT [90].

Overall, different somatostatin-based radiotracers have reached approval by regulatory agencies, and, currently, research in this field is very encouraging as to improving detection abilities and broadening potential clinical indications (Table 5) [17].

There is conflicting data about the influence of long-acting SSR inhibition in the performance of SSR-PET. Usually, it is recommended to do a wash out treatment period of four weeks, or to conduct imaging just prior to dosing with long-acting SST analogs. For patients with short-acting analogs, the recommendation is to stop treatment 24 h before [91].

These statements were made before the PET-CT era, and it is important to highlight that several trials with 68Ga-DOTATATE PET-CT showed no alteration in the uptake of the tumours or in the results. Chronic corticosterioid administration may influence SST2 expression, limiting the tumour detection ability of this tool. [92,93].

#### 5.1.2. The Role of PET-CT with 18F-FDG (Fluorodeoxyglucose)

Fluorodeoxyglucose is a glucose analogue transported into the cells by GLUT proteins. Once in the cell, it is phosphorylated and, after this process, glucose enters into the metabolic pathway of glycolysis, but FDG remains in the cell. Therefore, tumour cells have a higher metabolic activity with an abnormal higher uptake of FDG. FDG PET-CT is not widely used in well differentiated NETs; however, guidelines and different consensus are considering its use in G3 NETs presenting with negative 68Ga-DOTA-SSA and in cases with rapid progression regardless of tumour grade [94]. With a reported sensitivity of 40–66% in GEP-NENs, 18F FDG PET-CT has been used and patients with higher uptake in this technique had a lower overall survival [95].

The role of combining 68Ga-DOTA-SST analogues and 18F FDG PET-CT is under current research. A systematic review in this regard shows several advantages. It indicates that it is a feasible technique that may give the treating physician more information about the heterogeneity of the tumour, mainly by identifying clones with more aggressive features. The authors recommend the use of the combined technique at the time of diagnosis in patients with intermediate tumour proliferative activity (G2), if there is a different SSTR expression among different tumour lesions, and in non-functioning tumours, when patients have tumour-related symptoms. It may be used during follow-up, in addition to conventional imaging at the time of first disease restaging after changing systemic treatment, at the time of disease progression after prolonged stable disease, and in the case of discrepancy between conventional radiological evaluation and clinical/biochemical assessment [96,97].

## 6. Conclusions

Histopathological features, immunohistochemical profiles, and molecular differences allow clinicians to identify and establish prognosis and plan potential treatment alternatives. With more accurate imaging technology, improved staging, better surgical procedures, and smarter systemic treatments, patients will live longer and better. Our job as clinicians is to interpret all the information, apply it with each patient at the right time, and to integrate and articulate all the disciplines involved in the diagnosis and treatment of these patients. The disease burden, ECOG performance status, symptoms of the disease, disease volume, sites of metastases, histopathological characteristics, molecular information, and molecular imaging will help us in this decision-making process.

Integrating multiple disciplines and different diagnosis techniques in order to perform an adequate categorization of each patient will lead us to choose the optimal treatment sequence and to establish the best indication for local therapies in localized, locally advanced and metastatic disease.

Figure 3, Figure 4 and Figure 5 will help clinicians to integrate this information and to consider potential treatments.

## Figures and Tables

**Figure 1 cancers-14-02514-f001:**
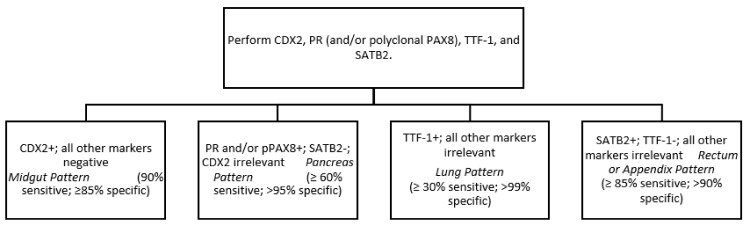
Simplified immunohistochemical algorithm for NETs site of origin by Bellizzi [27].

**Figure 2 cancers-14-02514-f002:**
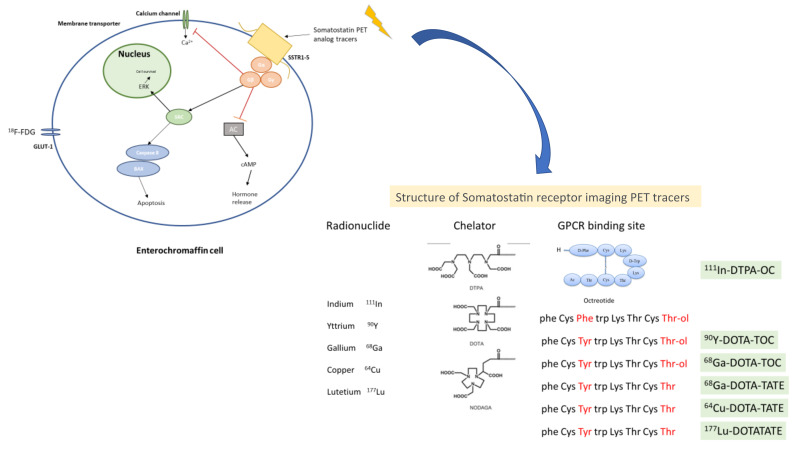
Structure and mechanism of action from somatostatin receptor imaging PET tracers. The Figure explains the basic molecular mechanisms of the endocrine cell and the main receptors used for diagnosis in NEN based on PET imaging. Legend: AC: adenylate cyclase, SSTR1–5: somatostatin receptor types 1–5, cAMP: cyclic AMP, SRC: proto-oncogene tyrosine-protein kinase, ERK: extracellular signal-regulated kinases, 123I-MIBG: metaiodobenzylguanidine, GLUT-1: glucose transporter 1, 18F-FDG: fluorodeoxyglucose.

**Figure 3 cancers-14-02514-f003:**
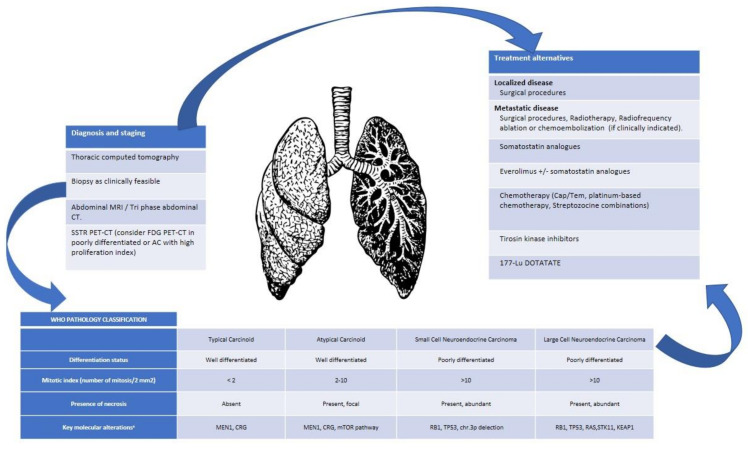
Integrative algorithm of current management in lung NENs. * No specific data for NET-G3 available. Cap/Tem, capecitabine/Temozolomide; chr., chromosome; CT, computed tomography; FDG, fluorodeoxyglucose; G, grade; IV, intravenous; LOH, loss of heterozygosity; MRI, magnetic resonance imaging; NEC, neuroendocrine carcinoma; NET, neuroendocrine tumour; PET-CT, positron emission tomography-computed tomography; sd., syndrome; SSTR, somatostatin receptors.

**Figure 4 cancers-14-02514-f004:**
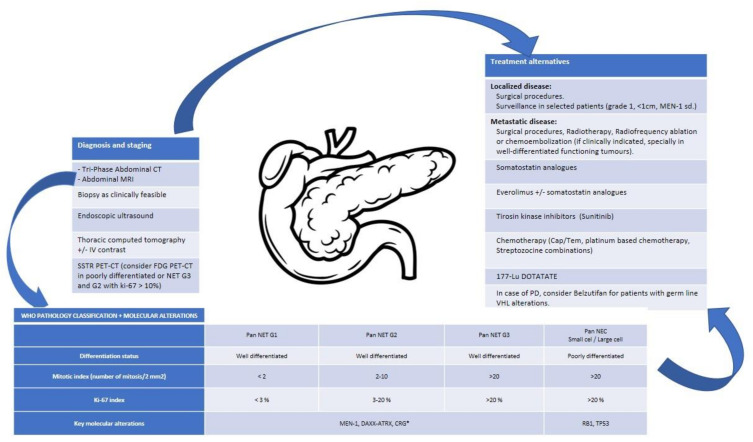
Integrative algorithm of current management in Pan NENs.* No specific data for NET-G3 available. Cap/Tem, capecitabine/Temozolomide; CRG, cromatin remodeling genes; CT, computed tomography; FDG, fluorodeoxyglucose; G, grade; IV, intravenous; MRI, magnetic resonance imaging; NEC, neuroendocrine carcinoma; NET, neuroendocrine tumour; Pan., pancreatic; PD, progressive disease; PET-CT, positron emission tomography-computed tomography; sd., syndrome; SSTR, somatostatin receptors; VHL, Von Hippel-Lindau [98].

**Figure 5 cancers-14-02514-f005:**
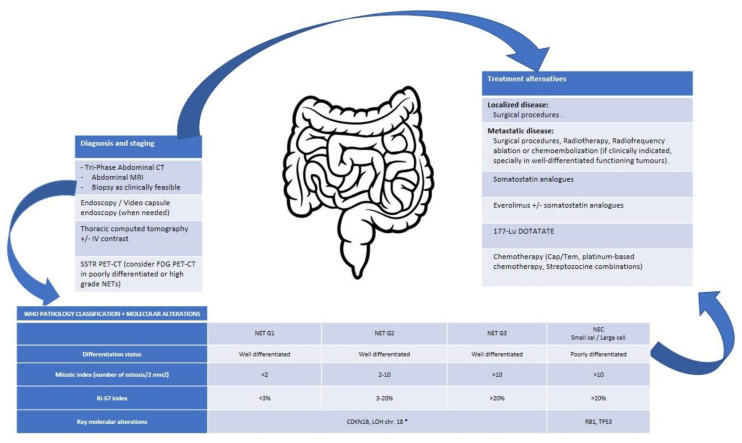
Integrative algorithm of current management in SI NENs.* No specific data for NET-G3 available. Cap/Tem, capecitabine/Temozolomide; chr., chromosome; CT, computed tomography; FDG, fluorodeoxyglucose; G, grade; IV, intravenous; LOH, loss of heterozygosity; MRI, magnetic resonance imaging; NEC, neuroendocrine carcinoma; NET, neuroendocrine tumour; PET-CT, positron emission tomography-computed tomography; sd., syndrome; SSTR, somatostatin receptors.

**Table 2 cancers-14-02514-t002:** Classification of digestive NENs (2019) [8].

	NET G1	NET G2	NET G3	SCNEC	LCNEC
Differentiation status	Well differentiated	Well differentiated	Well differentiated	Poorly differentiated	Poorly differentiated
Mitotic index (number of mitosis/2 mm2)	<2	2–20	>20	>20	>20
Ki-67 index Ki-67	<3%	3–20%	>20%	>20%	>20%

G, grade; NET, neuroendocrine tumour; SCNEC, small cell neuroendocrine carcinoma; LCNEC, large cell neuroendocrine carcinoma.

**Table 3 cancers-14-02514-t003:** Classification of lung NENs (2021) [9].

	TC	AC	SCNEC	LCNEC
Differentiation status	Well differentiated	Well differentiated	Poorly differentiated	Poorly differentiated
Mitotic index (number of mitosis/2 mm2)	<2	2–10	>10	>10
Presence of necrosis	Absent	Present, focal	Present, abundant	Present, abundant

AC, atypical carcinoid; TC, typical carcinoid; SCNEC, small cell neuroendocrine carcinoma; LCNEC, large cell neuroendocrine carcinoma.

**Table 4 cancers-14-02514-t004:** Most frequent hereditary syndromes in NENs.

Hereditary Syndrome	Gene (Location)	Protein	Clinical Features
MEN-1	MEN-1 (chr.11q13)	Menin	Parathyroid tumours, primary hyperparthyroidisms (95%), duodenopancratic NETs (40%), pituitary tumours (30%). Lung and thymus carcinoids, lipomas, and angiomas
MEN-2	RET (chr. 10)	RET	Medullary thyrioid carcinoma (>95%), pheochromocytomas (40–50%). MEN2A primary hyperparathyroidism (10–20%). MEN2B: ganglioneuromas
MEN-4	CDKN1B (chr. 12p13)	P27	Primary hyperparathyroidism, pituitari tumours, NET
VHL	VHL (chr.3p25)	pVHL	VHL type 1: retinal and CNS hemangioblastomas, lung, kidney and pancreatic cysts, clear renal cell carcinoma and pancreatic NETs (5–17%), VHL-2: adds pheochromocytomas (20%)
NF1	NF1 (chr.17q11.2)	Neurofibromin	“Café-au-lait” skin spot, neurofibromas and optic gliomas. Duodenopancretic NETs (1%) and pheohromocytomas (0,7%)
TSC	TSC1/TSC2 (chr.9q34/chr.16p13.3)	Harmartin, tuberin	Skin and neurological abnormalities, renal angiomyolipomas, hamartomas, functioning P-NETs

Chr, chromosome; CNS, central nervous system; NETs, neuroendocrine tumours; NF, neurofibromatosis; MEN, multiple endocrine neoplasia; TSC, tuberous sclerosis complex; VHL, Von Hippel–Lindau.

**Table 5 cancers-14-02514-t005:** Ligand-binding affinities according to in vitro assays from somatostatin-based radiochemicals (50% inhibitory concentration (IC50) in nM ± standard error of the mean) with clinical data.

	SST1	SST2	SST3	SST4	SST5	Cinical Development
**111 In-DTPA-OC**	>10,000	22 ± 3.6	182 ± 13	>1000	237 ± 52	FDA approved
**90Y-DOTA-TOC**	>10,000	11 ± 1.7	389 ± 135	>1000	114 ± 29	Phase II studies
**68Ga-DOTA-TOC**	>10,000	2.5 ± 0.5	613 ± 140	>1000	73 ± 2	EMA approved
**68Ga-DOTA-TATE**	>10,000	0.2 ± 0.04	>1000	300 ± 140	377 ± 18	FDA approved
**68Ga-DOTA-NOC**	>10,000	1,9 ± 0.4	40 ± 5.8	260 ± 74	7.2 ± 1.6	Phase II studies
**177Lu-DOTA-TATE**	>1000	2,0 ± 0.8	162 ± 16	>1000	>1000	FDA/EMA approved
**68Ga-DOTA-LM3**	-	12.5 ± 4.3	>1000	-	>1000	Prospective phase I/II
**68Ga-NODAGA-LM3**	-	1.3 ± 0.3	>1000	-	>1000	Prospective phase I/II
**68Ga-NODAGA-JR11 (Ga-OPS202)**	>1000	1.2 ± 0.2	>1000	>1000	>1000	Prospective phase I/II
**177Lu-DOTA-JR11 (Lu-OPS201)**	>1000	0.73 ± 0.15	>1000	>1000	>1000	Prospective phase I/II
**68Ga-DOTA-JR11 (Ga-OPS201)**	>1000	29 ± 2.7	>1000	>1000	>1000	Pilot study

DTPA: diethylenetriaminepentaacetic acid; DOTA: 1,4,7,10-tetraazacyclododecane-1,4,7,10-tetraacetic acid; NODAGA: 1,4,7-triazacyclononane,1-glutaric acid-4,7-acetic acid; FDA: Food and Drug Administration; EMA: European Medicines Agency.

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
