# Peer review of "Diagnosis in Neuroendocrine Neoplasms: From Molecular Biology to Molecular Imaging"

_cancers, 2022, doi:10.3390/cancers14102514_

Round 1

Reviewer 1 Report

Kopp et al. described in this review the diagnostic approach in neuroendocrine neoplasms (NENs), with a focus on molecular biology and imaging.

The topic is interesting, but several points need to be better developed.

Major Comments

  • The title of this review is “Diagnosis in neuroendocrine neoplasms: from molecular biology to molecular imaging”. On the basis of this title I expect to be included also liquid biopsy and NETtest, a clinical application of transcriptomic signature of NENs.
  • Molecular alterations: This aspect should be better developed for small cell lung cancer. Please, see and cite: PMID: 33331739
  • The diagnostic approach and related imaging techniques in functioning NENs are not reported (ectopic ACTH syndrome, insulinoma, glucagon-like peptide-1 receptor imaging, etc). Please see and cite: PMID: 26158607, PMID: 31951592
  • Endoscopic US is the most sensitive method to diagnose pancreatic NENs. This topic is not discussed in the “Imaging” paragraph.
  • Conclusions: a clearer message for the reader should be provided.

Minor Comments

  • Paragraph 3 Molecular alterations. Please, correct the number of the following paragraph and subparagraphs (2.1 Hereditary syndromes, etc… 3. Imaging in the diagnosis, staging and  follow  up  of  well-differentiated  neuroendocrine neoplasms ….)
  • Few typing errors in the text should be corrected.

Author Response

REVIEWER 1

Kopp et al. described in this review the diagnostic approach in neuroendocrine neoplasms (NENs), with a focus on molecular biology and imaging.

The topic is interesting, but several points need to be better developed.

Major Comments

  • The title of this review is “Diagnosis in neuroendocrine neoplasms: from molecular biology to molecular imaging”. On the basis of this title I expect to be included also liquid biopsy and NETtest, a clinical application of transcriptomic signature of NENs.

According to the reviewer´s suggestion we have included information regarding circulating biomarkers in NENs.

  • Molecular alterations: This aspect should be better developed for small cell lung cancer. Please, see and cite: PMID: 33331739

As suggested by the reviewer, additional details have been added related to SCLC (See 3.2 sporadic alterations: Lung NENs). Please note, that references have changed since new ones have been added.

“SCLC or SCLNEC is the most studied NEN of the lung. Globally, it is characterized by almost universal inactivation of TP53 and RB, the overexpression of oncogenes (Bcl-2 or MYC), and the alteration in key proliferation pathways such as PI3K/AKT/mTOR which is constitutively activated, Hedgehog or Notch. CNV are common in SCLC, the loss in chr.3p is present in around 90% of SCLC and it is considered to be present in pre-invasive lesions. Recently, Gay et al. described four SCLC subtypes characterized by the differential expression of three transcription factors ASCL1, NEUROD1, and POU2F3 or low expression of all the three with an inflamed gene signature (SCLC-A, N, P and I, respectively) with distinct response to systemic treatment”

  • The diagnostic approach and related imaging techniques in functioning NENs are not reported (ectopic ACTH syndrome, insulinoma, glucagon-like peptide-1 receptor imaging, etc). Please see and cite: PMID: 26158607, PMID: 31951592

According to reviewer´s comment, we have included additional information regarding imaging in functioning NENs.

  • Endoscopic US is the most sensitive method to diagnose pancreatic NENs. This topic is not discussed in the “Imaging” paragraph.

This topic is now included in the imaging section.

  • Conclusions: a clearer message for the reader should be provided.

We improved the conclusions and added key messages in each section. Also we integrated the information with figures 2,3,4 and 5.  

Minor Comments

  • Paragraph 3 Molecular alterations. Please, correct the number of the following paragraph and subparagraphs (2.1 Hereditary syndromes, etc… 3. Imaging in the diagnosis, staging and  follow  up  of  well-differentiated  neuroendocrine neoplasms ….)

As indicated, we have corrected the numbers of the paragraphs 3.1 Hereditary syndromes, 3.2 Sporadic alterations, 4. Imaging in the diagnosis, staging and follow up of well-differentiated neuroendocrine neoplasms, 4.1 molecular imagin ,5. Conclusions

  • Few typing errors in the text should be corrected.

Some typing errors have been identified and corrected along all the text.

Reviewer 2 Report

This manuscript describes the pathology, molecular biomarkers and novel radiotracer-based imaging findings of neuroendocrine neoplasms of digestive system and lung. This is an important topic, and the authors shows aboundant information of molecular biology and imaging tests. However, I think it is uneasy to integrate or utilize these information or findings into clinical practice. For better understanding, I would suggest the authors to add figures and comments below.

I would suggest that the authors to add representative figures of SSTR-PETs.

Also a summary figure of diagnosis (and treatment),including molecular biomarkers and molecular imaging, and prognostic or predictive role of these findings would be added.

Author Response

This manuscript describes the pathology, molecular biomarkers and novel radiotracer-based imaging findings of neuroendocrine neoplasms of digestive system and lung. This is an important topic, and the authors shows aboundant information of molecular biology and imaging tests. However, I think it is uneasy to integrate or utilize these information or findings into clinical practice. For better understanding, I would suggest the authors to add figures and comments below.

As suggested by the reviewer, and to emphasize the take home messages of each section, we have added extra “Key messages” with hyphens at the sections 1 (clinical presentation), 2 (pathological examination and classification), 3.2 sporadic alterations (Pancreatic NENS, SI-NENs, and lung NENs)

  1. Pathological examination and classification
  2. Clinical presentation

Key messages:

  • NENs are a very heterogeneous group of tumours with many different clinical presentations. Diagnosis of functioning NETs is a challenge and clinical suspicious is crucial.
  • Carcinoid syndrome is the most frequent hormonal syndrome. It is characterized mainly by diarrhea and flushing, but life threatening manifestations such as carcinoid heart disease or carcinoid crisis could appear.
  • NECs presentation use to be similar to other non-neuroendocrine malignancies.
  1. Pathological examination and classification

Key messages:

  • For the diagnosis of NETs and NECs the first step is to demonstrate the epithelial origin of the tumour by the expression of cytokeratines and the neuroendocrine nature by the expression of classical neuroendocrine markers (synaptophisin, chromogranin and/or CD56).
  • Immunohistochemistry together with morphology are key to establish the origin of the primary tumour.

3.2 Sporadic alterations

  • Pancreatic NENs

Key messages:

  • Sporadic P-NETs show a lower mutational burden and different mutated genes compared to classical PDAC.
  • Somatic mutations in MEN-1 and DAXX-ATRX are the most frequent in sporadic P-NETs, around 15% of sporadic P-NETs harbour mTOR pathway mutations. On the other hand RB and TP 53 are the most frequently altered genes in P-NECs.
  • Germline mutations have been described in 17% of sporadic P-NETs, further validation is needed.
  • Epigenetic mechanisms are of great interest in NETs, CpG island methylator phenotype has been described in around 83% of P-NETs.
  • Small intestine

Key messages:

  • SI-NETs show a low mutational burden and a few genes with recurrent somatic muations, being the most frequent mutations in CDKN1B (around 5-8% of SI-NETs).
  • CNV are frequent in SI-NETs. LOH of chr. 18 has frequently been described although its meaning is not yet understood.
  • Despite RB and TP53 are not frequently mutated in SI-NETs, they can be inactivated by other mechanisms.
  • Lung NENs

Key messages:

  • Lung carcinoids have a low mutational burden compared to LCNECs.
  • Chromatin-remodelling genes are the most frequently mutated both in NECs and lung carcinoids.
  • Mutations in MEN1 are the most frequent in low-grade carcinoids, whereas mutations in RB, TP53, PI3K/Akt/mTOR and RAS are more frequent in lung NECs.
  • Molecular study of lung NENs could potentially help to better classify these tumours helping to better understand their biology and prognosis.

* I would suggest that the authors to add representative figures of SSTR-PETs.

According to reviewer´s suggestion we have included a figure on SSTR-PETs.

* Also a summary figure of diagnosis (and treatment),including molecular biomarkers and molecular imaging, and prognostic or predictive role of these findings would be added.

We added 3 summary figures of diagnosis, classification and treatment alternatives that included molecular features for lung NENs, gastrointestinal and pancreatic NETs

Round 2

Reviewer 1 Report

The paper has reached a high enough priority to be acceptable for publication 

Reviewer 2 Report

The authors well revised the manuscript according to the reviewers comments, and have added important key messages and figures for better understanding.